# DISTILLED DIFFUSION LANGUAGE MODELS

## ABSTRACT

Transformer-based Large Language Models (LLMs) have demonstrated remarkable capabilities, yet their autoregressive nature forces sequential token-by-token decoding, leading to inefficiencies during inference. Furthermore, autoregressive language models lack inherent self-correction abilities, which hinders their capacity to refine and improve generated content without relying on external prompting or retraining techniques. In contrast, diffusion-based models offer the advantage of fast parallel generation through iterative refinement, while leveraging bi-directional attention to utilize full context at once. However, diffusion models are unable to match their autoregressive counterparts. This motivates us to explore the possibility of distilling a pre-trained autoregressive (AR) language model (teacher) into a non-autoregressive diffusion (non-AR) language model (student), combining the best of both worlds. In this work, we present *Target Concrete Score* (TCS) distillation, a theoretically grounded framework that bridges autoregressive and diffusion paradigms. TCS distillation is broadly applicable to both discrete and continuous diffusion models, with any pre-trained autoregressive teacher model. We propose techniques to make TCS distillation scalable and efficient for transformer-based models, and show how it can both improve pre-trained diffusion language models and also train new models from scratch. Through comprehensive experiments on language modeling tasks, we demonstrate the effectiveness of our proposed methods.

## 1 INTRODUCTION

Autoregressive (AR) architectures are the bread and butter for the modern revolution in Large Language Models (LLMs) (Brown et al., 2020; Touvron et al., 2023; Shoeybi et al., 2019). These models have shown amazing capabilities on a large variety of NLP tasks, but they still suffer from inefficient inference, hallucinations (Ji et al., 2023a; Zhang et al., 2023), overconfidence (Xiong et al., 2023), and "reversal curse" (Berglund et al.). These problems probably arise from their causal nature as they are learned in a left-to-right manner. First, the causal nature of AR models prevents them from generating tokens *in parallel*, unless specific multi-token-prediction training strategies has been applied (Gloeckle et al., 2024; Cai et al.). Second, they are *unable to undo actions* made earlier in generation easily. In some tasks, the ability to refine their generations, for example, through self-reflection (Ji et al., 2023b), or chain of thought (Wei et al., 2022) type approaches, can enhance the performance of autoregressive LLMs. However, this iterative inference process can be time consuming, because iterative improvement is performed by extending the autoregressive generation process, to mimic the "error-correction" training of these models.

Motivated by these limitations, the research community has attempted to use diffusion modeling techniques for language modeling. Diffusion models have been very successful in image generation (Ho et al., 2020; 2022; Rombach et al., 2021; Gu et al., 2023) using an implicit *progressive denoising technique* that is akin to self-refinement. The image generation models, which work in continuous spaces, start from Gaussian noise and progressively turn them into images, by iteratively cleaning up intermediate images that they generate. A large body of works has been developed which extend diffusion models to discrete space to match the

target domain of language (Lou et al., 2024; Campbell et al., 2022; Austin et al., 2021; Dieleman et al., 2022). These models generate text by starting with a categorical distribution that is easy to sample from and progressively turn them into sensible language; since these discrete diffusion models enable tokens to be generated *in parallel*, it leads to faster inference speed (higher bandwidth of generated tokens), especially for longer generations. Unfortunately, discrete diffusion models have been challenging to train and do not always achieve optimal fluency and performance. Research, such as the SEDD (Lou et al., 2024) and Plaid (Gulrajani & Hashimoto, 2023), suggests that they are approximately at the level of GPT-2, still lagging behind state-of-the-art AR LLMs.

Distillation has long been used to transfer knowledge from stronger models to weaker models (Hinton et al., 2015) because learning from a teacher can be more effective than learning from data distribution because the teacher can provide *distributional supervision* over the whole space, not just the observed data. In this paper we aim to distill a strong AR teacher into a diffusion language model[1]. Since the diffusion model student is a parallel generation model, while the teacher is an AR model, off the shelf distillation techniques do not apply. To address this gap, we made the following contributions in this work:

- We propose a target concrete score (TCS) distillation objective to bridge the gap between autoregressive teacher and non-autoregressive student, to combine the benefits of both worlds. We show the connection of gradient-informed estimation to the target score matching in continuous diffusion.

- We introduce methods to apply TCS to transformer-based language models, by proposed methods for efficient estimation of the target concrete score from AR teacher model. resulting a family of distillation methods called Distilled Diffusion Language Models (DDLM). To optimize the compute, we propose top-K and gradient-informed estimation techniques.

- Our proposed methods work for diffusion models that operate in discrete space (e.g. Lou et al. (2023)), and for those that map discrete tokens to continuous spaces and learn the model in continuous space (e.g. Gulrajani & Hashimoto (2023))

- We demonstrate through extensive experiments that the proposed methods achieve faster convergence, efficient parallel generation and lower perplexity and superior downstream reasoning and controlled generation task performance DDLM inherits the strengths of autoregressive models while bringing novel benefits such as iterative refinement during generation, which shines particularly in complex tasks like in-filling, arithmetic and arbitrary prompting.

## 2 PRELIMINARIES

**Notation** Let $\mathcal{X} = \{1, \ldots, V\}$ be the discrete data space, where $V = |\mathcal{X}|$ denotes the cardinality of $\mathcal{X}$, or the vocabulary size in language modeling. We use $x \in \mathcal{X}$ to denote a single discrete token, and $\mathbf{x} \triangleq [x^1, \ldots, x^L] \in \mathcal{X}^L$ to denote a finite sequence of discrete tokens, where $L$ is the sequence length. We use $\mathbf{x}^i \triangleq x^i \in \mathcal{X}$ to denote the $i$-th token in the sequence. For any data token $x \in \mathcal{X}$, denote $\mathbf{e}_x$ as the corresponding one-hot vector. For a sequence of tokens $\mathbf{x}$, we use $\mathbf{e_x} \in \mathbb{R}^{V \times L}$ to denote its one-hot representation $\mathbf{e_x} \triangleq [\mathbf{e}_{x^1}, \ldots, \mathbf{e}_{x^L}] \in \mathbb{R}^{V \times L}$. Given a matrix $\mathbf{M}$, we use $M_{ij}$ to denote the element at the $i$-th row and $j$-th column, and use $M_{i,:}$ and $M_{:,j}$ to represent the $i$-th row and $j$-th column, respectively. The identity matrix is denoted by $\mathbf{I}$. Throughout this paper, $q(\cdot)$ represents the distributions in forward process (adding noise), while $p(\cdot)$ denotes the distributions in reverse process (denoising). The base noise distribution is denoted as $p_T(\mathbf{x})$. We use $[\mathsf{M}] \in \mathcal{X}$ to denote the absorbing state in discrete diffusion model. We include a notation table for the distributions used in this paper in Table 2.

---

[1]Note that our method can actually use non-AR teachers as well, but not the focus in this paper.

**Discrete diffusion models: All you need is a good concrete score estimation** $\left[\frac{q_t(\hat{\mathbf{x}}_t)}{q_t(\mathbf{x}_t)}\right] \leftarrow \mathbf{s}_\theta(\mathbf{x}_t, t)$

FORWARD PROCESS  The forward process of a discrete diffusion model can be formulated as a continuous-time Markov chain (Campbell et al., 2022) (CTMC) $\{X_t\}_{t \in [0,T]}$, characterized by a rate matrix $\mathbf{R}_t \in \mathbb{R}^{V \times V}$, which satisfies $R_t(b, b) = -\sum_{a \neq b}^V R_t(b, a)$, and $R_t(a, b) \geq 0$ if $a \neq b$. In particular, the transition probability of the CTMC is $q_{t|0}(\mathbf{x}_t = b | \mathbf{x}_0 = a) = \left(\exp\left(\int_0^t R_s \mathrm{d}s\right)\right)_{ab}, a, b \in \mathcal{X}$. For a small $\Delta t \to 0$, it can be approximated using the Euler discretization $q_{t+\Delta t}(\mathbf{x}_{t+\Delta t} = b | \mathbf{x}_t = a) \approx \delta(b, a) + \Delta t R_t(b, a)$, with $\delta(b, a) = 1$ when $b = a$ and zero otherwise. By designing an appropriate rate matrix, one can transform a data distribution into a target distribution that is more accessible. For example, (Austin et al., 2021; Campbell et al., 2022; Sun et al., 2023) describe a diffusion with a uniform target and (Lou et al., 2024; Shi et al., 2024) model the rate matrix associated to an absorbing (masking) state

$$\mathbf{R}_t^{\mathrm{unif}} = \mathbf{1}\mathbf{1}^T - V\mathbf{I}, \quad \mathbf{R}_t^{\mathrm{mask}}(b, a) = \delta([\mathsf{M}], a) - \mathbf{I}_{ba}, \tag{1}$$

REVERSE PROCESS  Similar to continuous diffusion, discrete diffusion defined above has a time reversal governed by the reverse transition rate matrix

$$\overline{R}_t(\mathbf{x}_t, \hat{\mathbf{x}}_t) = \frac{q_t(\hat{\mathbf{x}}_t)}{q_t(\mathbf{x}_t)} R_t(\hat{\mathbf{x}}_t), \quad \hat{\mathbf{x}}_t \neq \mathbf{x}_t \tag{2}$$

where $q_t$ is the marginal distribution of $\mathbf{x}_t$ of the forward process. The intractable ratio $\frac{q_t(\hat{\mathbf{x}}_t)}{q_t(\mathbf{x}_t)}$ acts as an analog to the score function $\nabla_{\mathbf{x}_t} \log q_t(\mathbf{x}_t)$ in continuous diffusion as shown above. To estimate the intractable ratio, existing approaches resort estimating this density ratio with a neural network. Examples of such approaches include concrete score matching (Meng et al., 2022), categorical ratio matching (Sun et al., 2023), and denoising score entropy estimation (Lou et al., 2024). While these methods have achieved success in various applications, diffusion models are generally considered less effective than autoregressive models for language modeling.

## 3 TARGET CONCRETE SCORE DISTILLATION

In this work we focus on the problem of learning a diffusion model $p_\theta(\mathbf{x}_0)$ from a known distribution $q_0(\mathbf{x}_0)$. We depart from the standard diffusion setting whereby $p_\theta$ is trained with access to only samples from unknown data distribution $p_{\mathrm{data}}$. In our setting we explore the advantages of additionally having access to the true data distribution density $q_0$, as well its score $\nabla_{\mathbf{x}_0} \log q_0(\mathbf{x}_0)$, as in Bortoli et al. (2024).

We introduce *Target Concrete Score* (TCS) distillation as a general framework to make this possible. We first present the method in the context of discrete diffusion models, and then discuss its extension and connections to continuous diffusion models.

We assume access to a given pretrained autoregressive model $q_{\mathrm{AR}}(\mathbf{x}) = \prod_{l=1}^L q_{\mathrm{AR}}(x^l | \mathbf{x}^{<l})$, for $\mathbf{x} \in \mathcal{X}, x^l \in \mathcal{V}$ as our target distribution. $\mathbf{x}^{<l}$ represents a vector containing the variables from $x^1$ up to and including $x^{l-1}$, while $\mathbf{x}^{>l}$ is similarly defined for variables with index greater than $l$. Note that the proposed TCS distillation is applicable to any known distribution $q_0$, but we limit our scope to an autoregressive density estimator given the potential benefits of parallel sampling, as discussed in Section 1.

Given $q_0(\mathbf{x}_0) \triangleq q_{\mathrm{AR}}(\mathbf{x}_0)$, we construct a probability path with the marginal distribution $q_t(\mathbf{x}_t) = \sum_{\mathbf{x}_0} q_{t|0}(\mathbf{x}_t | \mathbf{x}_0) q_0(\mathbf{x}_0)$. As introduced in Section 2, the reverse process can be described by the backward rate matrix $\overline{R}_t(\mathbf{x}, \hat{\mathbf{x}})$ (Campbell et al., 2022, Prop. 1), which has the following form :

$$\overline{R}_t(\mathbf{x}, \hat{\mathbf{x}}) = R_t(\hat{\mathbf{x}}, \mathbf{x}) \sum_{\mathbf{x}_0} \frac{q_{t|0}(\hat{\mathbf{x}} | \mathbf{x}_0)}{q_{t|0}(\mathbf{x} | \mathbf{x}_0)} q_{0|t}(\mathbf{x}_0 | \mathbf{x}). \tag{3}$$

Notice that the forward process conditional $q_{t|0}$ is known and tractable, while the time-reversal conditional $q_{0|t}$ is unknown and intractable. Thus, to recover the exact time-reversal of the defined forward process induced by $q_{\mathrm{AR}}$ and $q_{t|0}$, we use a parametric denoising model $p_{0|t}(\mathbf{x}_0|\mathbf{x}_t; \theta) \triangleq p_\theta(\mathbf{x}_0|\mathbf{x}_t)$ to approximate the time-reversal conditional $q_{0|t}(\mathbf{x}_0|\mathbf{x}_t)$. This can be achieved by minimizing the following objective:

$$\mathcal{J}(\theta; w(\cdot), \mathbb{D}(\cdot \| \cdot)) := \int_0^T w(t)\mathbb{E}_{q(\mathbf{x}_t)}\mathbb{D}\left(q_{0|t}(\mathbf{x}_0|\mathbf{x}_t) \| p_\theta(\mathbf{x}_0|\mathbf{x}_t)\right)\mathrm{d}t, \tag{4}$$

where $w : [0, T] \to \mathbb{R}_{>0}$ is a positive weighting function and $\mathbb{D}(\cdot\|\cdot)$ is a discrepancy measure between two distributions. Note that this objective resembles the score matching objective (Song et al., 2021) in continuous diffusion models, which shares essentially the same goal of matching the score of the forward marginal distribution $q_t(\mathbf{x}_t)$.

After training $\theta^* = \arg\min_\theta \mathcal{J}(\theta)$, the backward rate matrix $\bar{\mathbf{R}}$ can be computed by replacing $q_{0|t} \approx p_{\theta^*}(\mathbf{x}_0|\mathbf{x}_t)$ in Equation (2). Samples can then be drawn by simulating the backward CTMC using Euler discretization as described in Section 2. To optimize the objective in Equation (4), we should specify the discrepancy measure $\mathbb{D}$.

**Remark 1.** *One option is the Kullback-Leibler (KL) divergence, which gives us the objective resemble the maximum likelihood $\mathcal{L}_{\mathrm{KL}}(\theta) = -\mathbb{E}_{t\sim U(0,1)}\mathbb{E}_{q_{\mathrm{AR}}(\mathbf{x}_0)q_{t|0}(\mathbf{x}_t|\mathbf{x}_0)}[\log p_\theta(\mathbf{x}_0|\mathbf{x}_t)] + C$ where $C$ denotes a constant independent of $\theta$.*

We propose *Target Concrete Score* (TCS) distillation, an effective approach to train a diffusion model $p_\theta(\mathbf{x}_0)$ by distilling a pretrained autoregressive language model $q_{\mathrm{AR}}(\mathbf{x}_0)$. We resort to matching the concrete score (Meng et al., 2022) in Equation (4). To be precise, given a distribution $p(\mathbf{x})$, we define the log-density ratio vector of a token at the $l$-th position to be $\mathbf{r}_{p(\mathbf{x})}(x^l) \in \mathbb{R}^{V\times 1}$ with $\mathbf{r}_{p(\mathbf{x})}(x^l) = \left[\log \frac{p(\mathbf{x}^{<l},x',\mathbf{x}^{>l})}{p(\mathbf{x}^{<l},x^l,\mathbf{x}^{>l})}\right]_{x'\in\mathcal{V}}$. Similarly, we define the log-density ratio matrix $\mathbf{r}_{p(\mathbf{x})}(\mathbf{x}) \in \mathbb{R}^{V\times L}$ for distribution $p(\mathbf{x})$ evaluated at $\mathbf{x} = \left[x^1, \ldots, x^L\right]$ as follows:

$$\mathbf{r}_{p(\mathbf{x})}(\mathbf{x}) \triangleq \left[\mathbf{r}_{p(\mathbf{x})}(x^1) \quad \cdots \quad \mathbf{r}_{p(\mathbf{x})}(x^l) \quad \cdots \quad \mathbf{r}_{p(\mathbf{x})}(x^L)\right] \in \mathbb{R}^{V\times L}. \tag{5}$$

We can relate the defined log-density ratio matrix $\mathbf{r}_{p(\mathbf{x})}(\mathbf{x})$ to the concrete score[2] $\mathbf{c}_{p(\mathbf{x})}(\mathbf{x})$ for distribution $p(\mathbf{x})$ evaluated at $\mathbf{x}$ by

$$\mathbf{c}_{p(\mathbf{x})}(\mathbf{x}; \mathcal{N}) \triangleq \exp\left[\mathbf{r}_{p(\mathbf{x})}(x^1), \ldots, \mathbf{r}_{p(\mathbf{x})}(x^l), \ldots, \mathbf{r}_{p(\mathbf{x})}(x^L)\right]. \tag{6}$$

where we define the neighbors set $\mathcal{N}(\mathbf{x}) \triangleq \{\mathbf{y} \mid \mathbf{y} \in \mathcal{X}^L, \text{Hamming distance}(\mathbf{x}, \mathbf{y}) = 1\}$ and the $\exp$ function is applied element-wise to the matrix. Analogous to score matching in continuous domains, we can utilize such concrete score-based discrepancy measure to quantify the difference between two discrete probability distributions. This concept is formally stated in the following proposition:

**Proposition 1.** *(Meng et al., 2022) Let $p(\mathbf{x})$ and $q(\mathbf{x})$ be two distributions over the discrete support $\mathcal{X}^L$, $\mathbf{c}_{p(\mathbf{x})}(\mathbf{x}; \mathcal{N}) = \mathbf{c}_{q(\mathbf{x})}(\mathbf{x}; \mathcal{N})$, or equivalently $\mathbf{r}_{p(\mathbf{x})}(\mathbf{x}) = \mathbf{r}_{q(\mathbf{x})}(\mathbf{x})$, implies that $p(\mathbf{x}) = q(\mathbf{x}), \forall\mathbf{x} \in \mathcal{X}^L$.*

Therefore, we can align the concrete scores of the student and teacher models to minimize the objective in Equation (4), leading to the target concrete score distillation objective

$$\mathcal{J}(\theta; w(\cdot)) := \int_0^T w(t)\mathbb{E}_{q(\mathbf{x}_0,\mathbf{x}_t)}\mathcal{D}(\mathbf{r}_{q_{0|t}(\mathbf{x}_0|\mathbf{x}_t)}(\mathbf{x}_0), \mathbf{r}_{p_\theta(\mathbf{x}_0|\mathbf{x}_t)}(\mathbf{x}_0))\mathrm{d}t. \tag{7}$$

---

[2] Note that the concrete score defined in this paper differs from that in Meng et al. (2022), though they are equivalent up to a constant. Specifically, the relationship is given by $\mathbf{c}^{\mathrm{Meng}}(\mathbf{x}) = \mathbf{c}^{\mathrm{Ours}}(\mathbf{x}) - 1$.

where $\mathcal{D} : \mathbb{R}^{V \times L} \times \mathbb{R}^{V \times L} \to \mathbb{R}$ represents a general loss function that measures the discrepancy between two matrices. This can include various forms such as distance metrics or divergence measures.

To minimize the objective, it requires an estimation of the log-density ratio of $q_{0|t}$, which is

$$\log \frac{q_{0|t}(\widehat{\mathbf{x}}_0|\mathbf{x}_t)}{q_{0|t}(\mathbf{x}_0|\mathbf{x}_t)} = \log \frac{q_{\mathrm{AR}}(\widehat{\mathbf{x}}_0)}{q_{\mathrm{AR}}(\mathbf{x}_0)} + \log \frac{q_{t|0}(\mathbf{x}_t|\widehat{\mathbf{x}}_0)}{q_{t|0}(\mathbf{x}_t|\mathbf{x}_0)}. \tag{8}$$

Thanks to the tractability of $q_{\mathrm{AR}}$, both terms $\log q_{\mathrm{AR}}(\mathbf{x}_0) = \sum_{l=1}^{L} \log q_{\mathrm{AR}}(x_0^l|\mathbf{x}^{<l})$ and $\log q_{t|0}(\mathbf{x}_t|\mathbf{x}_0)$ are known and tractable. This gives us a tractable form of the target concrete score distillation objective:

---

**Target Concrete Score (TCS) Distillation Objective**

$$\mathcal{J}_{\mathsf{TCS}}(\theta; w(\cdot)) := \int_0^T w(t) \mathbb{E}_{q(\mathbf{x}_0, \mathbf{x}_t)} \mathcal{D}(\underbrace{\mathbf{r}_{q_{\mathrm{AR}}(\mathbf{x}_0)}(\mathbf{x}_0) + \mathbf{r}_{q_{t|0}(\mathbf{x}_t|\mathbf{x}_0)}(\mathbf{x}_0)}_{\text{Teacher}}, \; \underbrace{\mathbf{r}_{p_\theta(\mathbf{x}_0|\mathbf{x}_t)}(\mathbf{x}_0)}_{\text{Student}}) \mathrm{d}t. \tag{9}$$

---

**Remark 2.** *When the forward process is associated with the masking rate matrix $\mathbf{R}_t^{\mathrm{mask}}$, we have $q_{t|0}(\mathbf{x}_t|\mathbf{x}_0) = q_{t|0}(\mathbf{x}_t|\widehat{\mathbf{x}}_0)$ (Shi et al., 2024; Sahoo et al., 2024), which implies $\mathbf{r}_{q_{0|t}(\mathbf{x}_0|\mathbf{x}_t)}(\mathbf{x}_0) = \mathbf{r}_{q_{\mathrm{AR}}(\mathbf{x}_0)}(\mathbf{x}_0)$. Consequently, the TCS distillation objective can be further simplified as*

$$\mathcal{J}_{TCS}^{\mathrm{mask}}(\theta; w(\cdot)) := \int_0^T w(t) \mathbb{E}_{q(\mathbf{x}_0, \mathbf{x}_t)} \mathcal{D}(\mathbf{r}_{q_{\mathrm{AR}}(\mathbf{x}_0)}(\mathbf{x}_0), \; \mathbf{r}_{p_\theta(\mathbf{x}_0|\mathbf{x}_t)}(\mathbf{x}_0)) \mathrm{d}t. \tag{10}$$

### 3.1 MODEL PARAMETERIZATION

We have previously introduced the TCS distillation objective in Equation (9) for the general discrete diffusion case. However, we have not yet discussed how to parameterize the concrete score of the denoising model distribution $\mathbf{r}_{p_\theta(\mathbf{x}_0|\mathbf{x}_t)}(\mathbf{x}_0)$ in detail, which will be addressed in this section.

**Concrete Score Parameterization** $\mathbf{r}_{p_\theta(\mathbf{x}_0|\mathbf{x}_t)}(\mathbf{x}_0) \triangleq \mathbf{s}_\theta(\mathbf{x}_t, t) \in \mathbb{R}^{V \times L}$ Similar to Lou et al. (2024), we can use a neural network $\mathbf{s}_\theta(\mathbf{x}_t, t)$ to approximate the target concrete score $\mathbf{c}_{q_{0|t}(\mathbf{x}_0|\mathbf{x}_t)}(\mathbf{x}_0) \triangleq \exp\left[\mathbf{r}_{q_{\mathrm{AR}}(\mathbf{x}_0)}(\mathbf{x}_0)\right]$. Particularly, we can use the score entropy loss function used by Lou et al. (2024) as the discrepancy measure $\mathcal{D}(\cdot, \cdot)$ in Equation (9), where $\mathcal{D} = \mathcal{D}_F\left(\mathbf{s}_\theta(\mathbf{x}_t, t), \exp\left[\mathbf{r}_{q_{\mathrm{AR}}(\mathbf{x}_0)}(\mathbf{x}_0)\right]\right)$ is the Bregman divergence $\mathcal{D}_F(p, q) = F(p) - F(q) - \langle \nabla F(q), p - q \rangle$. with convex function $F = -\log$. This gives us the following TCS objective with score parameterization:

$$\mathcal{J}_{\mathsf{TCS}}(\theta; w(\cdot)) := \int_0^T w(t) \mathbb{E}_{q(\mathbf{x}_0, \mathbf{x}_t)} \mathcal{D}_F(\exp\left[\mathbf{r}_{q_{\mathrm{AR}}(\mathbf{x}_0)}(\mathbf{x}_0)\right], \; \mathbf{s}_\theta(\mathbf{x}_t, t)) \mathrm{d}t. \tag{11}$$

**Denoising Mean Parameterization** $p_\theta(\mathbf{x}_0|\mathbf{x}_t) = \prod_{l=1}^{L} \mathrm{Cat}(x_0^l; \mathrm{softmax}\left[\boldsymbol{\mu}_\theta(\mathbf{x}_t, t)\right]_{:,l})$ Similar to Campbell et al. (2022); Shi et al. (2024), we can directly parameterize the denoising distribution $p_\theta(\mathbf{x}_0|\mathbf{x}_t)$ by a neural network $\boldsymbol{\mu}_\theta(\mathbf{x}_t, t) \in \mathbb{R}^{V \times L}$ which outputs the logits of the categorical distribution at each position.

With this factorized parameterization, matching the concrete score between $q_{0|t}(\mathbf{x}_0|\mathbf{x}_t)$ and $p_\theta(\mathbf{x}_0|\mathbf{x}_t)$ is equivalent to matching the concrete score at each position, which leads us to the following objective based on cross-entropy minimization

$$\mathcal{J}_{\mathsf{TCS}}(\theta; w(\cdot)) := \int_0^T w(t) \mathbb{E}_{q(\mathbf{x}_0, \mathbf{x}_t)} \sum_{l=1}^{L} \mathbb{H}\left(\mathrm{Cat}\left(x_0^l; \mathrm{softmax}\left[\mathbf{r}_{q_{0|t}}(x_0^l|\mathbf{x}_t)\right]\right), \; p_\theta(x_0^l|\mathbf{x}_t)\right) \mathrm{d}t. \tag{12}$$

# 4 DISTILLED DIFFUSION LANGUAGE MODELS

In this section, we demonstrate how to apply the TCS objective to a specific setup of interest: distilling a pre-trained transformer-based autoregressive language model $q_{\mathrm{AR}}$ to a denoising diffusion language model $p_\theta$. We present a set of techniques to facilitate the efficient computation of the target concrete score $\mathbf{r}_{q_{\mathrm{AR}}}(\mathbf{x}_0)$ in practice for transformer-based language models. We refer to the family of models resulting from this process as **D**istilled **D**iffusion **L**anguage **M**odels, or DDLM.

## 4.1 EFFICIENT ESTIMATION OF TARGET CONCRETE SCORE

To optimize the TCS distillation objective, we need to compute the target concrete score $\mathbf{r}_{q_{\mathrm{AR}}(\mathbf{x}_0)}(\mathbf{x}_0)$. Naively, this requires $(V-1) \times L + 1$ log-density evaluations of the teacher autoregressive model for each sequence $\mathbf{x}$, where for each position $1 \leq l \leq L$, the $l$-th token is replaced with all other $V-1$ tokens, and the log probability of each altered sequence is explicitly computed by the teacher model to obtain the log-density ratio, ultimately resulting in the target concrete score $\mathbf{r}_{q_{\mathrm{AR}}}(\mathbf{x}_0)$. However, this procedure is computationally prohibitive. To address this challenge, we propose two practical estimation approaches. For example, GPT-2 (Radford et al., 2019) has 50257 vocabulary size and Llama3 (Dubey et al., 2024) model has 128_000 vocabulary size. We introduce two approaches to efficiently estimate the target concrete score, *top-$K$* estimation and *gradient-informed* estimation.

**Top-$K$ Estimation.** Empirically, the concrete score is highly sparse. As illustrated in Figure 2, tokens with high density ratios closely resemble the one-hot encoding of original tokens in the simplex space, but enriched with distributional information. This observation motivates approximating the score vector with only the top-$K$ items, treating the rest as zero, for efficient computation. In particular, we approximate the computation of $\mathbf{r}_{q_{\mathrm{AR}}(\mathbf{x}_0)}(\mathbf{x}_0)$ by replacing the $l$-th token only with the top-$K$ most probable tokens, determined by the logits output of teacher model based on the preceding $l-1$ tokens, estimated by the teacher model itself $q_{\mathrm{AR}}(x^l | \mathbf{x}^{<l})$. This approach reduces the total number of sequence log-probability evaluations from $(V-1) \times L$ to $K \times L + 1$, thus eliminating the dependency on vocabulary size. Note that we can read out $q_{\mathrm{AR}}(x^l | \mathbf{x}^{<l})$ from the teacher model's logits output at each position $l$, which can be done in one batched forward pass with causal attention. Additionally, we employ KV-caching during the teacher model's forward pass to further reduce computational overhead. The details of the top-$K$ estimation algorithm is described in Algorithm 2. We found this approach to be effective in practice with a relatively small $K \leq 128$.

**Gradient-informed Estimation.** We now present another method to estimate the target concrete score $\mathbf{r}_{q_{\mathrm{AR}}(\mathbf{x}_0)}(\mathbf{x}_0)$. The key insight is that while autoregressive language models operate over discrete state spaces, they are, in fact, continuous and differentiable functions that accept real-valued one-hot encoded input tokens, though they are typically evaluated on a discrete subset of their domain. This observation has been employed in previous work to accelerate the convergence rate of Gibbs sampling in discrete energy-based models (Grathwohl et al., 2021).

To compute the log-density ratio $\log \frac{q_{\mathrm{AR}}(\hat{\mathbf{x}}_0)}{q_{\mathrm{AR}}(\mathbf{x}_0)}$, we can use the first-order Taylor approximation to estimate it $\log \frac{q_0(\hat{\mathbf{x}}_0)}{q_0(\mathbf{x}_0)} = \nabla_{\mathbf{x}_0} \log q_0(\mathbf{x}_0)^\top (\hat{\mathbf{x}}_0 - \mathbf{x}_0) + o(\|\hat{\mathbf{x}}_0 - \mathbf{x}_0\|) \approx \nabla_{\mathbf{x}_0} \log q_0(\mathbf{x}_0)^\top (\hat{\mathbf{x}}_0 - \mathbf{x}_0)$ Note that by the definition $\hat{\mathbf{x}}_0$ and $\mathbf{x}_0$ only differ in one position, we can estimate the concrete score efficiently $\hat{\mathbf{r}}_{q_{\mathrm{AR}}(\mathbf{x}_0)}(\mathbf{x}_0)_{ij} = \nabla_{\mathbf{x}_0} \log q_0(\mathbf{x}_0; \phi)_{ij} - \mathbf{e}_{x_0^j}^\top \nabla_{\mathbf{x}_0} \log q_0(\mathbf{x}_0)_{:,j}$, where $\hat{\mathbf{r}}_{q_{\mathrm{AR}}(\mathbf{x}_0)}(\mathbf{x}_0)_{ij}$ approximates log-probability ratio at replacing the $j$-th token of sentence $\mathbf{x}_0$ with the $i$-th token in the vocabulary $\mathcal{V}$. Compared to the exact computation, such gradient-based estimation $\hat{\mathbf{r}}$ involves just one forward and backward pass to evaluate the log-probability of the teacher model and one backward pass to obtain its gradient, significantly reducing the computational cost. For further details, see the pseudo-code in Listing 1.

**Algorithm 1** Baseline Diffusion LM Training (Left) & DDLM Training (Right)

| | |
|---|---|
| 1: **procedure** LM-BATCH($\mathbf{x}_0$, model, optim) | 1: **procedure** DDLM-BATCH($\mathbf{x}_0$, model, teacher, optim, $L, V, K$) |
| 2:     $\mathbf{x}_0 \leftarrow$ one_hot($\mathbf{x}$) | 2:     $\mathbf{x}_0 \leftarrow$ one_hot($\mathbf{x}$) |
| 3:     targets $\leftarrow$ one_hot($\mathbf{x}$) | 3:     targets $\leftarrow$ one_hot($\mathbf{x}$) |
| 4:     $\mathbf{x}_t \sim q_{t\|0}(\mathbf{x}_t\|\mathbf{x}_0)$ | 4:     tcs $\leftarrow$ tcs_estimate($\mathbf{x}_0$, teacher, $L, V, K$) |
| 5:     logits $\leftarrow$ model($\mathbf{x}_t$) | 5:     tcs_targets $\leftarrow$ softmax(tcs), dim$=-1$) |
| 6:     loss $\leftarrow$ CE(logits, targets) | 6:     $\mathbf{x}_t \sim q_{t\|0}(\mathbf{x}_t\|\mathbf{x}_0)$ |
| 7:     loss.backward(); optim.step() | 7:     logits $\leftarrow$ model($\mathbf{x}_t$) |
| 8: **end procedure** | 8:     loss $\leftarrow \lambda$CE(logits, targets) + CE(logits, tcs_targets) |
| | 9:     loss.backward(); optim.step() |
| | 10: **end procedure** |

## 4.2 TCS DISTILLATION FOR CONTINUOUS DIFFUSION LANGUAGE MODELS

Our DDLM is a versatile distillation framework that can be easily extended to not only discrete target distributions but also continuous ones. To see this, we define the forward process of continuous diffusion models as $q_{t|0}(\mathbf{z}_t|\mathbf{x}_0) = \prod_{l=1}^{L} q_{t|0}(z_t^l|x_0^l)$ and $q_{t|0}(z_t|x_0) = \mathcal{N}(z_t; \alpha_t \mathbf{E}^\top \mathbf{e}_{x_0}, \sigma_t^2 \mathbf{I})$,[3] where $\mathbf{E} \in \mathbb{R}^{V \times d}$ denotes the word embedding matrix, $\mathbf{e}_{x_0}$ is the one-hot representation of the token $x_0 \in \mathcal{V}$. The diffusion model can be parametrize as a denoising prediction $p_\theta(\mathbf{x}_0|\mathbf{z}_t)$. To learn the student $p_\theta$ through distillation from the teacher $q_{\mathrm{AR}}$, we can apply the objective in Remark 1, which straightforwardly extends to continuous scenarios as the same objective applies. Similarly, the TCS objective in Equation (9) remains valid since the posterior $q_{0|t}(\mathbf{x}_0|\mathbf{z}_t)$ is discrete, and Proposition 1 still holds. To estimate the concrete score, we can employ both top-$K$ and gradient-based estimation. Moreover, we can establish the connection between our TCS objective Equation (9) and target score matching (Bortoli et al., 2024) (TSM) is proposed for continuous diffusion models, as introduced below and detailed in Appendix A.

**Proposition 2.** *Target score matching objective above is equivalent to a first-order Taylor approximation of our TCS objective.*

## 4.3 DDLM TRAINING ALGORITHM

Building on the TCS objective introduced in Equation (9) and the two practical estimation methods discussed earlier, we present the full training procedure for DDLM, as illustrated in Algorithm 1. In TCS distillation, data examples must be sampled from the target teacher distribution. However, relying exclusively on teacher-policy data for distillation may not yield optimal results. When autoregressive LLMs are trained using the teacher-forcing objective, their learned distributions can become biased and skewed, potentially resulting in less diverse and artificially generated data samples. Alternatively, when real data is available, it can be sampled and evaluated by the teacher model to compute the target concrete score. Indeed, the TCS distillation objective is also effective for any $\mathbf{x}_0 \sim q_0(\mathbf{x}_0)$ with full support over $\mathcal{X}^L$, enabling off-policy data learning. In practice, to balance data efficiency with sample quality, we sample from a mixture of teacher-generated data and real data: $\mathbf{x}_0 \sim \omega q_{\mathrm{AR}}(\mathbf{x}_0) + (1-\omega)q_{\mathrm{data}}(\mathbf{x}_0)$, with the default value of $\omega = 0.5$. Similar to the classical knowledge distillation (Hinton et al., 2015), we combine the TCS distillation loss with the denoising score matching loss of the baseline student model as a weighted sum controlled by $\lambda$ as shown in Algorithm 1.

---

[3]Rather than using $\mathbf{x}_t$, here we denote the latent variable of diffusion models as $\mathbf{z}_t = [z_t^1, \ldots, z_t^L], z^l \in \mathbb{R}^d$, to emphasize that it lies in a continuous space.

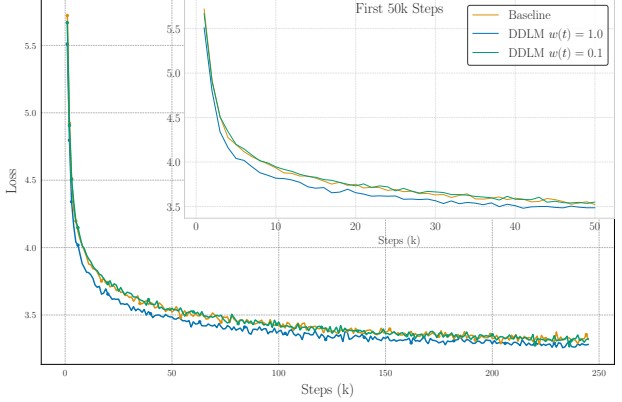

| Autoregressive | |
| --- | --- |
| Transformer-XL | 23.5 |
| *Discrete Diffusion - Uniform* | |
| SEDD Uniform | $\leq 40.25$ |
| DDLM Student SEDD Uniform | |
| *Discrete Diffusion - Absorb* | |
| SEDD Absorb (33B tokens) | $\leq 32.79$ |
| DDLM Student SEDD Absorb | |
| *Autoregressive (Retrained)* | |
| Transformer (33B tokens) | 22.32 |
| MDLM (33B tokens) | $\leq 27.04$ |
| MDLM (327B tokens) | $\leq 23.00$ |
| *Discrete Diffusion* | |
| DDLM  AR Teacher(327B) | 20.86 |
| DDLM Student MDLM (33B tokens) | $\leq 24.2$ |
| DDLM Student MDLM (327B tokens) | $\leq 22.1$ |

Figure 1: Progression of validation negative log-likelihood (NLL) loss on the OPENWEBTEXT dataset during training. The inset magnifies the first 50,000 steps for clarity.

Table 1: Test perplexities (PPL ↓) on LM1B dataset.

## 5 EXPERIMENTS

In this section, we empirically assess the performance of DDLM with TCS distillation across various language modeling and reasoning tasks to investigate the following research questions: **RQ1:** Is TCS distillation an effective training objective for distilling a pre-trained autoregressive (AR) language model into a diffusion language model? **RQ2:** Does such distillation offer novel benefits for AR language modeling? **RQ3:** What are the limitations of the proposed TCS distillation? Under what conditions does TCS distillation perform best, and when does it fall short? We present a summary of our findings in this section. Detailed descriptions of the datasets and model configurations can be found in the appendix due to space constraints.

**Baselines** We use state-of-the-art diffusion language models in both discrete and continuous settings as the baseline models, including SEDD (Shi et al., 2024), MD4 (Shi et al., 2024), MDLM (Sahoo et al., 2024) in discte space and Plaid (Gulrajani & Hashimoto, 2023) in continuous space.

**DDLM Models** In our experiments, we consider the following DDLM models: **DDLM-Full** refers to the model that uses the exact TCS estimation computed by replacing each token with all other tokens in the vocabulary. This is possible when the vocabulary size $V$ is small such as character-level language modeling tasks. **DDLM-TopK** refers to the model that uses the top-$K$ approximation of the TCS estimation. **DDLM-$\nabla$** refers to the model that uses the gradient-based estimation of the TCS. We include the name of the base student model in the name of the DDLM model for clarity, e.g., DDLM-Student-SEDD meaning that we use SEDD as the diffusion language model formulation to distill the teacher AR model. We use DDLM-from-scratch to refer to the model that is trained from scratch, and DDLM-fine-tune to refer to the model that the student model is first pre-trained by regular denoising score matching objective and then fine-tuned by our TCS distillation.

> **Summary of Findings I**
>
> TCS distillation in DDLM significantly and consistently enhances the learning efficiency of student diffusion language models.

LANGUAGE MODELING We conducted experiments in language modeling using the OPENWEBTEXT dataset. Initially, we pre-trained a transformer-based autoregressive model with the same configurations as in (Sahoo et al., 2024). We employed the absorbing discrete diffusion model (Sahoo et al., 2024; Shi et al., 2024) as our base student model. Utilizing DDLM with Top-K estimation where $K = 128$, we trained the model from scratch. We experimented with various weighting schemes for the TCS objective, ranging from 0.01 to 1.0,

and compared the results with a baseline model that did not use TCS distillation. We plot the validation negative log-likelihood (NLL) loss on the OPENWEBTEXT dataset in Figure 1. The results indicate that TCS distillation indeed accelerates the learning process of the student model. Additionally, we observed that the distillation loss consistently resulted in lower perplexity compared to the baseline throughout the training.

We also present perplexity results on the LM1B dataset in Table 1. For this, we pre-trained an AR teacher model from (Sahoo et al., 2024) and applied DDLM-TopK with $K = 128$. We experimented with different backbone models for the student model, including SEDD and MDLM. Our findings show that, with the same number of training tokens, the distilled student model outperforms the baseline SEDD and MDLM models.

REASONING We also test the reasoning ability of the distilled student model following the setting in (Deng et al., 2023; Ye et al., 2024). We follow the same training recipe in (Ye et al., 2024) to fine-tune the AR model the augmented GSM8K dataset, as well as training the diffusion language model for the task. Figure 3 illustrates the validation accuracy on the GSM8K-Aug dataset during training, comparing an autoregressive (AR) fine-tuned model and DDLM against a teacher model benchmark. The DDLM demonstrates superior performance, achieving faster initial learning and higher overall accuracy compared to the fine-tuned AR model. This performance difference highlights the DDLM's efficiency in convergence and generalization, making it a preferable choice for tasks that require rapid and effective learning.

> **Summary of Findings II**
>
> DDLM can unlock new capabilities for teacher model into distilled student model.

**DDLM enables faster parallel generation** DISCRETE DIFFUSION We employed GPT2-Medium as our teacher model and used DDLM-Top-$K$ for distillation. For the student model, we utilized GPT2-Small with an absorbing discrete diffusion model. Unlike previous language modeling experiments, we solely used the data generated by the teacher model to distill the student model. We conducted experiments with both DDLM-from-scratch and DDLM-fine-tune approaches. Our findings indicate that we can retain approximately 3% of the original performance in terms of generative perplexity, as evaluated by GPT2-Large, while achieving at least a 3x speedup in generation.

CONTINUOUS DIFFUSION Parallel generation can be pushed even further by using continuous Gaussian diffusion models, where advanced samplers (Lu et al., 2022) and ODE solvers (Karras et al., 2022) can be readily applied in straight-forward manner. To test the limit of this approach, we re-train the Plaid model (Gulrajani & Hashimoto, 2023) using the GPT2 tokenizer, and apply DDLM-TopK to distill from the AR teacher model, which is GPT2-Medium. We show the results in Figure 5, where

**In-filling, Arbitrary Prompting, and Controlled Generation** As shown in SEDD paper (Lou et al., 2024), the concrete score formulation of discrete diffusion model naturally extends to in-filling, arbitrary prompting, and controlled generation tasks. Based on the established framework, we further combine it with DDLM fine-tuning to enable controlled generation via external constraints. Here we consider a toy task following the work (Hu et al., 2023). The task is to prompt the language model to generate random numbers from a given distribution, different from the work (Hu et al., 2023) which uses autoregressive style left to right prompt: "The following is a random single-digit integer drawn uniformly between 0 and 9:". Here our diffusion language model student allows us to prompt the model in arbitrary order. We format the prompt as: "The single-digit integer [M] is uniformly drawn between 0 and 9.". In this controlled generation task, the constraint can be formulated as $p_{\text{constraint}}(x) \propto \delta(x \in \{0, \dots, 9\})$. We can either apply the constraint during the sampling process in a training-free manner, or via DDLM-fine-tuning, where the previous one is garantted to work. The DDLM-fine-tuning results are presented in Figure 4, along with the results from the AR teacher model. It's evident that for both causal left-to-right prediction and the estimated target concrete score, the autoregressive (AR) teacher model displays a highly biased and skewed distribution. By employing DDLM to distill this into a diffusion model, we can achieve controlled generation in a much more straightforward manner.

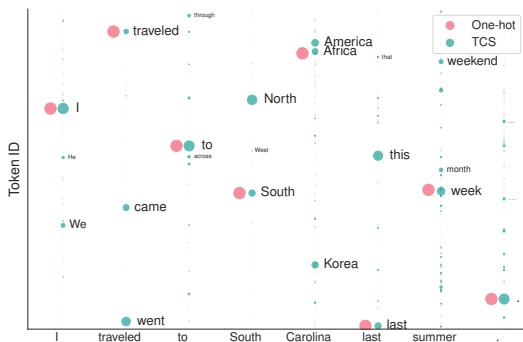

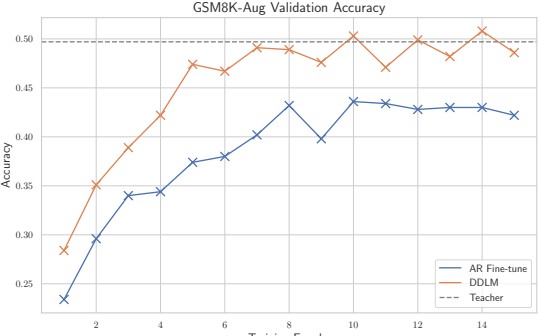

Figure 2: Estimated target concrete score at each token position.

Figure 3: Validation accuracy on GSM8K-Aug dataset during training.

> **Summary of Findings III**
>
> DDLM-fine-tuning with TCS distillation can transfer the reasoning ability of the teacher model to the student model. The distilled student model can maintain the teacher model's reasoning capability without requiring additional intermediate reasoning tokens. Thanks to the parallel sampling process, the distilled student model can reason more efficiently.

Following the methodology outlined by (Ye et al., 2024), we evaluated the reasoning capability of the distilled student model on the multi-digit multiplication task from the BIG-bench benchmark (Srivastava et al., 2022), which is considered the most challenging among arithmetic tasks. Specifically, we focused on four-digit (4 x 4) and five-digit (5 x 5) multiplication problems, as these tasks are particularly difficult to solve without using Chain of Thought (CoT) reasoning. We employed a fine-tuned AR model as the teacher model and tested the distilled student model on these tasks. The results are presented in Table 3. Our findings indicate that the DDLM-fine-tune approach can achieve comparable or even better results than the baseline, relying solely on the supervision provided by the teacher model.

> **Summary of Findings IV**
>
> While DDLM-from-scratch and DDLM-fine-tune can improve the sample efficiency of the student model, they do not always improve the final task performance, particularly with the presence of extensive data augmentation and amount of data.

We observe that the benefits of DDLM-fine-tune are task-dependent. Specifically, DDLM-fine-tune does not consistently enhance the student model's performance in terms of perplexity for language modeling tasks. It is crucial to use ground-truth data during fine-tuning to maintain the teacher model's perplexity. However, DDLM-fine-tune can yield better results in terms of generative perplexity. Additionally, we note that when the dataset is large and data augmentation is extensive, as in the case of GSM8K-Aug, the distillation benefits may plateau.

## 6 CONCLUSION

In this work, we propose a novel framework for distilling pre-trained autoregressive models into denoising diffusion language models. We proposed a novel target concrete score (TCS) distillation objective, along with DDLM models for transformer-based language models. Extensive experiments on language modeling tasks demonstrate the effectiveness of the proposed framework.

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

# Appendix

CONTENTS

| Notation | Description |
|---|---|
| $q_0(\mathbf{x}_0)$ | Teacher model distribution |
| $q_{t\|0}(\mathbf{x}_t\|\mathbf{x}_0)$ | Forward process (adding noise) |
| $q_{0,t}(\mathbf{x}_0, \mathbf{x}_t) = q_{t\|0}(\mathbf{x}_t\|\mathbf{x}_0)q_0(\mathbf{x}_0)$ | Joint distribution at time $t$ |
| $q_t(\mathbf{x}_t) = \sum_{\mathbf{x}_0} q_{t\|0}(\mathbf{x}_t\|\mathbf{x}_0)q_0(\mathbf{x}_0)$ | Marginal distribution at time $t$ |
| $q_{0\|t}(\mathbf{x}_0\|\mathbf{x}_t) = \frac{q_{t\|0}(\mathbf{x}_t\|\mathbf{x}_0)q_0(\mathbf{x}_0)}{q_t(\mathbf{x}_t)}$ | Time-reversal conditional distribution at time $t$ |

Table 2: Notations and their descriptions

## A CONNECTION TO TARGET SCORE MATCHING

In this section, we establish the connection between the proposed target concrete score distillation objective and the original target score matching objective (Bortoli et al., 2024). We begin by introducing target score matching, which serves as an objective for training a distilled diffusion model. We then demonstrate its equivalence to our target concrete score distillation objective under specific assumptions.

Recall that in continuous diffusion language models, the forward process is defined as $q_{t|0}(\mathbf{z}_t|\mathbf{x}_0) = \prod_{l=1}^{L} q_{t|0}(\mathbf{z}_t^l|\mathbf{e}_{x_0^l}) = \prod_{l=1}^{L} \mathcal{N}(\mathbf{z}_t^l; \alpha_t \mathbf{E}^\top \mathbf{e}_{x_0^l}, \sigma_t^2 \mathbf{I})$. To learn a diffusion model through knowledge distillation, we can parameterize its score function using a neural network $\mathbf{s}_\theta(\mathbf{z}_t, t)$, which is trained to approximate the true score $\nabla_{\mathbf{z}_t} \log q_t(\mathbf{z}_t)$. To achieve this, we employ target score matching (Bortoli et al., 2024). Specifically, we present the following Lemma.

**Lemma 1** (Target Score Matching Identity). *Let $p(\mathbf{z}_t|\mathbf{x}_0) = \mathcal{N}(\mathbf{z}_t; \alpha_t \mathbf{x}_0, \sigma_t^2 \mathbf{I})$ and $p(\mathbf{x}_0)$ be any differentiable distribution. We have the identity*

$$\nabla_{\mathbf{z}_t} \log p(\mathbf{z}_t) = \frac{1}{\alpha_t} \mathbb{E}_{p(\mathbf{x}_0|\mathbf{z}_t)} \left[ \nabla_{\mathbf{x}_0} \log p(\mathbf{x}_0) \right]. \tag{13}$$

*Proof.* The proof follows that in Bortoli et al. (2024) with the generalization to the scaled Gaussian convolutions. Specifically, using the translation-invariant property of Gaussian distribution, we obtain $\nabla_{\mathbf{x}_0} \log p(\mathbf{z}_t|\mathbf{x}_0) = -\alpha_t \nabla_{\mathbf{z}_t} \log p(\mathbf{z}_t|\mathbf{x}_0)$. Applying Bayes' rule, we then have:

$$\nabla_{\mathbf{z}_t} \log p(\mathbf{z}_t|\mathbf{x}_0) = -\frac{1}{\alpha_t} \nabla_{\mathbf{x}_0} \log p(\mathbf{z}_t|\mathbf{x}_0)$$

$$= -\frac{1}{\alpha_t} \nabla_{\mathbf{x}_0} \log p(\mathbf{x}_0|\mathbf{z}_t) + \frac{1}{\alpha_t} \nabla_{\mathbf{x}_0} \log p(\mathbf{x}_0).$$

Together with the denoising score identity, we have

$$\nabla_{\mathbf{z}_t} \log p(\mathbf{z}_t) = \int p(\mathbf{x}_0|\mathbf{z}_t)\nabla_{\mathbf{z}_t} \log p(\mathbf{z}_t|\mathbf{x}_0)\mathrm{d}\mathbf{x}_0$$

$$= \frac{1}{\alpha_t} \int p(\mathbf{x}_0|\mathbf{z}_t)\Big(-\nabla_{\mathbf{x}_0} \log p(\mathbf{x}_0|\mathbf{z}_t) + \nabla_{\mathbf{x}_0} \log p(\mathbf{x}_0)\Big)\mathrm{d}\mathbf{x}_0$$

$$= \frac{1}{\alpha_t} \int \nabla_{\mathbf{x}_0} p(\mathbf{x}_0|\mathbf{z}_t) \log p(\mathbf{x}_0)\mathrm{d}\mathbf{x}_0,$$

where the last equality holds since $\mathbb{E}_{p(\mathbf{x}_0|\mathbf{z}_t)}\nabla_{\mathbf{x}_0} \log p(\mathbf{x}_0|\mathbf{z}_t) = 0$. □

Using Lemma 1, the score neural network can be learned by minimizing the target score matching loss

$$\mathcal{L}_{\mathrm{TSM}}(\theta) = \mathbb{E}_{t\sim U(0,1)}\mathbb{E}_{q_0(\mathbf{x}_0)q_{t|0}(\mathbf{z}_t|\mathbf{x}_0)} \left\|\mathsf{s}_\theta(\mathbf{z}_t, t) - \frac{1}{\alpha_t}\nabla_{\mathbf{x}_0} \log p(\mathbf{x}_0)\right\|_2^2. \qquad (14)$$

To draw a connection to the proposed TCS objective, we utilize the mean prediction parametrization $\boldsymbol{\mu}_\theta(\mathbf{z}_t, t) \approx \mathbb{E}_{q_{0|t}(\mathbf{x}_0|\mathbf{z}_t)}[\mathbf{x}_0]$ instead. Using Tweedie's formula $\mathbb{E}_{q_{0|t}(\mathbf{x}_0|\mathbf{z}_t)}[\mathbf{x}_0] = \frac{1}{\alpha}(\sigma_t^2\nabla_{\mathbf{z}_t} \log q_t(\mathbf{z}_t) + \mathbf{z}_t)$ and rescaling it by the signal-noise ratio $\lambda_t \triangleq \frac{\alpha_t^2}{\sigma_t^2}$, we can reparametrize $\mathcal{L}_{\mathrm{TSM}}$ as

$$\arg\min_\theta \mathcal{L}_{\mathrm{TSM}}(\theta) \Leftrightarrow \arg\min_\theta \mathbb{E}_{t\sim U(0,1)}\mathbb{E}_{q_0(\mathbf{x}_0)q_{t|0}(\mathbf{z}_t|\mathbf{x}_0)} \left\|\boldsymbol{\mu}_\theta(\mathbf{z}_t, t) - \left(\nabla_{\mathbf{x}_0} \log p_0(\mathbf{x}_0) + \frac{\alpha_t}{\sigma_t^2}\mathbf{z}_t\right)\right\|_2^2. \quad (15)$$

In optimal training, we have $\boldsymbol{\mu}_{\theta^*}(\mathbf{z}_t, t) = \frac{1}{\lambda_t}\mathbb{E}_{q_{0|t}(\mathbf{x}_0|\mathbf{z}_t)}[\mathbf{x}_0]$. Then, we are ready to prove Proposition 2.

**Proposition 2.** *Target score matching objective above is equivalent to a first-order Taylor approximation of our TCS objective.*

*Proof.* Consider the log-probability ratio $\log \frac{q_{0|t}(\hat{\mathbf{x}}_0|\mathbf{z}_t)}{q_{0|t}(\mathbf{x}_0|\mathbf{z}_t)}$, in which $\hat{\mathbf{x}}_0$ only differs $\mathbf{x}_0$ in the $i$-th position with $\hat{x}_0^i \neq x_0^i$. By applying the Bayes' rule, we have

$$\log \frac{q_{0|t}(\hat{\mathbf{x}}_0|\mathbf{z}_t)}{q_{0|t}(\mathbf{x}_0|\mathbf{z}_t)} = \log \frac{q_0(\hat{\mathbf{x}}_0)}{q_0(\mathbf{x}_0)} + \log \frac{q_{0|t}(\mathbf{z}_t|\hat{\mathbf{x}}_0)}{q_{0|t}(\mathbf{z}_t|\mathbf{x}_0)}.$$

The second term in RHS can be further simplified as

$$\log \frac{q_{0|t}(\mathbf{z}_t|\hat{\mathbf{x}}_0)}{q_{0|t}(\mathbf{z}_t|\mathbf{x}_0)} = \sum_{l=1}^L \log \frac{q_{t|0}(\mathbf{z}_t^l|\mathbf{e}_{\hat{x}_0^l})}{q_{t|0}(\mathbf{z}_t^l|\mathbf{e}_{x_0^l})} = \log \frac{q_{t|0}(\mathbf{z}_t^i|\mathbf{e}_{\hat{x}_0^i})}{q_{t|0}(\mathbf{z}_t^i|\mathbf{e}_{x_0^i})}$$

because $\hat{\mathbf{x}}_0$ and $\mathbf{x}_0$ only differ at the $i$-th position. Assume the word embedding matrix $\mathbf{E}$ used in the forward process is the identity matrix, then

$$\log q_{t|0}(\mathbf{z}_t^i|\mathbf{e}_{x_0^i}) \propto -\frac{\|\mathbf{z}_t^i - \alpha_t\mathbf{e}_{x_0^i}\|^2}{2\sigma_t^2} = -\frac{1}{2\sigma_t^2}\left[\|\mathbf{z}_t^i\|^2 - 2\alpha_t\langle\mathbf{z}_t^i, \mathbf{e}_{x_0^i}\rangle + \alpha_t^2\|\mathbf{e}_{x_0^i}\|^2\right]$$

$$\log \frac{q_{t|0}(\mathbf{z}_t^i|\mathbf{e}_{\hat{x}_0^i})}{q_{t|0}(\mathbf{z}_t^i|\mathbf{e}_{x_0^i})} = \frac{\alpha_t}{\sigma_t^2}\langle\mathbf{z}_t^i, \mathbf{e}_{\hat{x}_0^i} - \mathbf{e}_{x_0^i}\rangle - \frac{\alpha_t^2}{2\sigma_t^2}(\|\mathbf{e}_{\hat{x}_0^i}\|^2 - \|\mathbf{e}_{x_0^i}\|^2)$$

Since both $\mathbf{e}_{\hat{x}_0^i}$ and $\mathbf{e}_{x_0^i}$ are one-hot encoding, we can simplify the above term as

$$\log \frac{q_{t|0}(\mathbf{z}_t^i|\mathbf{e}_{\hat{x}_0^i})}{q_{t|0}(\mathbf{z}_t^i|\mathbf{e}_{x_0^i})} = \frac{\alpha_t}{\sigma_t^2}\langle\mathbf{z}_t^i, \mathbf{e}_{\hat{x}_0^i} - \mathbf{e}_{x_0^i}\rangle$$

For the marginal log-density ratio at $t = 0$, we estimate it using Taylor approximation, which gives

$$\log \frac{q_0(\hat{\mathbf{x}}_0)}{q_0(\mathbf{x}_0)} \approx \langle \nabla_{\mathbf{x}_0} \log q_0(\mathbf{x}_0), \mathbf{e}_{\hat{\mathbf{x}}_0} - \mathbf{e}_{\mathbf{x}_0} \rangle.$$

Combine above two results, we get

$$\log \frac{q_{0|t}(\hat{\mathbf{x}}_0|\mathbf{z}_t)}{q_{0|t}(\mathbf{x}_0|\mathbf{z}_t)} \approx \langle \nabla_{\mathbf{x}_0} \log q_0(\mathbf{x}_0), \mathbf{e}_{\hat{\mathbf{x}}_0} - \mathbf{e}_{\mathbf{x}_0} \rangle + \frac{\alpha_t}{\sigma_t^2} \langle \mathbf{z}_t^i, \mathbf{e}_{\hat{x}_0^i} - \mathbf{e}_{x_0^i} \rangle$$

Thus, the TCS target is

$$\mathbf{r}_{q_{0|t}}(\mathbf{x}_0|\mathbf{z}_t)_{i,j} = \log \frac{q_{0|t}(\hat{\mathbf{x}}_0|\mathbf{z}_t)}{q_{0|t}(\mathbf{x}_0|\mathbf{z}_t)} \approx \langle \nabla_{\mathbf{x}_0} \log q_0(\mathbf{x}_0), \mathbf{e}_{\hat{\mathbf{x}}_0} - \mathbf{e}_{\mathbf{x}_0} \rangle + \frac{\alpha_t}{\sigma_t^2} \langle \mathbf{z}_t^i, \mathbf{e}_{\hat{x}_0^i} - \mathbf{e}_{x_0^i} \rangle$$

Written in column vector form, this yields:

$$\mathbf{r}_{q_{0|t}}(\mathbf{x}_0|\mathbf{z}_t)_{:,i} = \left[ \langle \nabla_{\mathbf{x}_0} \log q_0(\mathbf{x}_0), \mathbf{e}_{\mathbf{x}_0|x_0^i \leftarrow j} - \mathbf{e}_{\mathbf{x}_0} \rangle + \frac{\alpha_t}{\sigma_t^2} \langle \mathbf{z}_t^i, \mathbf{e}_j - \mathbf{e}_{x_0^i} \rangle \right]_{j=1}^V$$

where $\mathbf{x}_0|x_0^i \leftarrow j \triangleq [x_0^1, \ldots, x_0^i = j, \ldots, x_0^L]$. By taking softmax operator on both sides, we have

$$\mathrm{softmax}(\mathbf{r}_{q_{0|t}}(\mathbf{x}_0|\mathbf{z}_t)_{:,i}) = \mathrm{softmax}\left( \left[ \langle (\nabla_{\mathbf{x}_0} \log q_0(\mathbf{x}_0))_{:,i}, \mathbf{e}_j \rangle + \frac{\alpha_t}{\sigma_t^2} \langle \mathbf{z}_t^i, \mathbf{e}_j \rangle \right]_{j=1}^V \right)$$

Written this in the matrix form, it yields:

$$\mathrm{softmax}(\mathbf{r}_{q_{0|t}}(\mathbf{x}_0|\mathbf{z}_t)) = \mathrm{softmax}\left( \nabla_{\mathbf{x}_0} \log p_0(\mathbf{x}_0) + \frac{\alpha_t}{\sigma_t^2} \mathbf{z}_t \right).$$

Using the denoising mean parametrization $p_\theta(\mathbf{x}_0|\mathbf{z}_t) = \prod_{l=1}^L \mathrm{Cat}(x_0^l; \mathrm{softmax}\,[\boldsymbol{\mu}_\theta(\mathbf{z}_t, t)]_{:,l})$ as mentioned in Section 3.1, we can learn the neural network $\boldsymbol{\mu}_\theta(\mathbf{x}_t, t)$ by minimizing the loss

$$\mathcal{L}(\theta) = \mathbb{E}_{t \sim U(0,1)} \mathbb{E}_{q_0(\mathbf{x}_0) q_{t|0}(\mathbf{z}_t|\mathbf{x}_0)} \left\| \boldsymbol{\mu}_\theta(\mathbf{z}_t, t) - \left( \nabla_{\mathbf{x}_0} \log p_0(\mathbf{x}_0) + \frac{\alpha_t}{\sigma_t^2} \mathbf{z}_t \right) \right\|_2^2, \tag{16}$$

which is identical to the original TSM loss defined in Equation (15). $\square$

## B  ALGORITM AND PSEUDO CODE

In this section, we present the pseudo-code for the proposed estimation methods of the target concrete score.

Algorithm 2 demonstrate the proposed Top-$K$ estimation. First, we compute the teacher model's logit output based on the preceding tokens. This can be achieved in a single forward pass using causal attention calculated in parallel. Next, the top-$K$ tokens at each position are selected to compute the log-density ratio, ultimately leading to the estimated concrete score.

We also introduce a variant called Top-$K$ with $N$-Gram estimation. In Algorithm 3, we highlight the differences in blue. This variant employs a distinct procedure for selecting the top tokens. At the $l$-th position, we use an N-Gram language model to compute n-gram scores and select additional top-$K$ tokens based on these scores, combining them with the original top-$K$ tokens selected from the teacher's logit. This results in a total of $2K$ tokens. Specifically, the n-gram score at position $l$ is computed as $[p(x^{l+1}, \ldots, x^{l+N-1}|x)]_{x \in \mathcal{V}}$ with $p(x^{l+1}, \ldots, x^{l+N-1}|x) \propto p(x)p(x, x^{l+1}, \ldots, x^{l+N-1})$, where $p(x)$ and $p(x, x^{l+1}, \ldots, x^{l+N-1})$ can be estimated using the empirical distribution. Empirically, we observe that this approach performs similarly to that in Algorithm 2.

We also provide the pseudo-code in Listing 1 for gradient-informed estimation. In this method, we use a first-order Taylor approximation to estimate the concrete score, significantly reducing computational costs.

---

**Algorithm 2** DDLM Top-K Estimation

1: **procedure** tcs_estimate($\mathbf{x}_0$, teacher_model, $L, V, K$, tcs)
2:          $\triangleright$ $\mathbf{x}_0$: Input tokens; $L$: Sequence length; $V$: Vocabulary size; $K$: Top-$K$ tokens to select; tcs: list
3:     logits $\leftarrow$ teacher_model($\mathbf{x}_0$) $\in \mathbb{R}^{V \times L}$; original_log_prob $\leftarrow$ teacher_model_log_prob($\mathbf{x}_0$)
4:     **for** $l = 1$ to $L$ **do**
5:        Get top-$K$ tokens: top_tokens $\leftarrow$ TopK(logits[:, $l$], $K$)
6:        If $\mathbf{x}_0[l] \notin$ top_tokens, add it to top_tokens
7:        Construct a batch of new sequences $\widehat{\mathbf{x}}_0 \leftarrow [\mathbf{x}_0^{<l}, \text{top\_tokens}, \mathbf{x}_0^{>l}]$
8:        Compute log probability of sequences log_prob from new_logits $\leftarrow$ teacher_model($\widehat{\mathbf{x}}_0$)
9:        Compute log-density ratio: log_density_ratio $\leftarrow$ log_prob $-$ orig_log_prob
10:       Append log-density ratio to list: tcs $\leftarrow$ tcs $+$ log_density_ratio
11:     **end for**
12:     **return** tcs
13: **end procedure**

---

**Algorithm 3** DDLM Top-K with N-Gram Estimation

1: **procedure** tcs_estimate($\mathbf{x}_0$, teacher_model, ngram_model, $L, V, K$, tcs)
2:          $\triangleright$ $\mathbf{x}_0$: Input tokens; $L$: Sequence length; $V$: Vocabulary size; $K$: Top-$K$ tokens to select; tcs: list
3:     logits $\leftarrow$ teacher_model($\mathbf{x}_0$) $\in \mathbb{R}^{V \times L}$; original_log_prob $\leftarrow$ teacher_model_log_prob($\mathbf{x}_0$)
4:     **for** $l = 1$ to $L$ **do**
5:        Get top-$K$ tokens: top_tokens $\leftarrow$ TopK(logits[:, $l$], $K$)
6:        Get N-Gram score for all tokens: n-gram_scores $\leftarrow$ ngram_model($[\mathbf{x}_0^{l+1}, \ldots, \mathbf{x}_0^{l+N-1}]$)
7:        Add another top-$K$ tokens: top_tokens $\leftarrow$ top_tokens $+$ TopK(n-gram_scores, $K$)
8:        If $\mathbf{x}_0[l] \notin$ top_tokens, add it to top_tokens
9:        Construct a batch of new sequences $\widehat{\mathbf{x}}_0 \leftarrow [\mathbf{x}_0^{<l}, \text{top\_tokens}, \mathbf{x}_0^{>l}]$
10:       Compute log probability of sequences log_prob from new_logits $\leftarrow$ teacher_model($\widehat{\mathbf{x}}_0$)
11:       Compute log-density ratio: log_density_ratio $\leftarrow$ log_prob $-$ orig_log_prob
12:       Append log-density ratio to list: tcs $\leftarrow$ tcs $+$ log_density_ratio
13:     **end for**
14:     **return** tcs
15: **end procedure**

---

Listing 1: Concrete Score Estimation with first-order Taylor approximation

```python
import torch
import torch.nn.functional as F

def ddlm_target_score_distillation(teacher_model, tokens, vocab_size,
    temperature=1.0):
  B, L = tokens.shape
  x_0 = F.one_hot(tokens.long(), num_classes=vocab_size).to(torch.float)
  with torch.enable_grad():
    x_0.requires_grad_(True)
    logits = teacher_model(x_0)
    log_prob = F.log_softmax(logits, dim=-1)
    log_prob = (x_0[:, 1:, :] * log_prob[:, :-1, :]).sum()
    log_prob.backward()
    grad_log_prob = x_0.grad
  # Compute log-density ratios
```

```
15      log_prob_ratio = grad_log_prob - (x_0 * grad_log_prob).sum(dim=-1,
            keepdim=True)
16
17      # Temperature adjustment
18      log_prob_ratio /= temperature
19
20      prob_ratio = torch.exp(log_prob_ratio)
21      return prob_ratio
```

## C  DETAILS OF EXPERIMENT

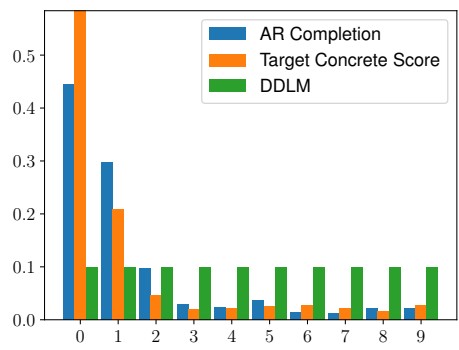

Figure 4: Distribution comparison when prompted with generating a random single-digit number between 0 and 0.

Figure 5: Parallel generation capabilities of DDLM

We first present experimental results of datasets: TEXT8 and LM1B, OPENWEBTEXT, and include a detailed analysis and summarization of our finding at the end of this ection.

We use the following datasets in our experiments. We use the same model configuration, training setup, and optimization hyperparameters as the corresponding baseline student models.

**TEXT8**  TEXT8 is a character-level text dataset consisting of a small vocabulary of 27 tokens: the letters a-z and the _ whitespace token. We follow the convention of training and evaluating text8 in chunks of length 256 without any preprocessing(Hoogeboom et al., 2021). We used the standard bits-per-character metric (BPC) for this dataset. Due to small vocabulary size 27, we can use DDLM-Full which uses teacher AR model to compute the exact target concrete score by replacing each token with all other tokens in the vocabulary.

**LM1B**  We also evaluate DDLM on the One Billion Words dataset, which is of medium size and represents real-world data. We adhere to the tokenization, training, and model size configurations outlined in (He et al., 2023). Specifically, our baseline models are approximately the size of GPT-2 small. Consistent with (He et al., 2023), we primarily compare against other language diffusion models, although we also train a standard autoregressive transformer for benchmarking purposes.

**OPENWEBTEXT** We follow (Lou et al., 2024) to test the language modeling capabilities of our model. We use the same training, validation, and test splits as in (Lou et al., 2024). We use batch size of $512$ and sequence length of $1024$ for training. We keep our evaluation setup the same as (Lou et al., 2024).

| Model | $4 \times 4$ | $5 \times 5$ | GSM8K-Aug |
|---|---|---|---|
| **No CoT** | | | |
| GPT-2 S | 0.29 | 0.01 | 0.13 |
| GPT-2 M | 0.76 | 0.02 | 0.17 |
| GPT-2 L | 0.34 | 0.01 | 0.13 |
| **Implicit CoT** | | | |
| GPT-2 S | 0.97 | 0.10 | 0.20 |
| GPT-2 M | 0.96 | 0.96 | 0.22 |
| **Explicit CoT** | | | |
| GPT-2 S | 1.00 | 1.00 | 0.41 |
| GPT-2 M | 1.00 | 1.00 | 0.44 |
| GPT-2 L | 1.00 | 0.99 | 0.45 |
| DoT Plaid | 1.00 | 1.00 | 0.33 |
| DDLM Plaid | 1.00 | 1.00 | 0.34 |

Table 3: Main results. Accuracy for multiplication tasks and GSM8K-Aug.

## D  RELATED WORKS

Diffusion models (Austin et al., 2021; Campbell et al., 2022; Sahoo et al., 2024; Lou et al., 2024; Campbell et al., 2024; Gat et al., 2024; Sun et al., 2023; Shi et al., 2024; He et al., 2023; Ye et al., 2023), grounded in discrete-time Markov chains within continuous state spaces and employing Gaussian transitions (Sohl-Dickstein et al., 2015; Ho et al., 2020), have been extended to continuous-time formulations through the application of stochastic processes and score matching (Song et al., 2021). A parallel research direction explores discrete diffusion models operating on discrete data spaces, similarly based on Markov chains (Sohl-Dickstein et al., 2015; Hoogeboom et al., 2021). D3PM (Austin et al., 2021) investigated discrete-time Markov chains utilizing various transition matrices (uniform, absorbing, discretized Gaussian), deriving a discrete-time variational lower bound (ELBO) that was subsequently generalized to continuous-time Markov chains (CTMCs) (Campbell et al., 2022). This approach leverages mean-parameterization to learn the reverse transition probability.

An alternative perspective posits that D3PM implicitly learns the ratio of marginal distributions, termed the "concrete score"—a discrete analogue of the continuous score function (Meng et al., 2022; Lou et al., 2024). This ratio can be directly learned via concrete score matching, mirroring the continuous score matching approach (Meng et al., 2022). However, practical implementation faces challenges due to the incompatibility of the L2 loss with the inherent positivity constraint of this ratio. SEDD (Lou et al., 2024) addresses this challenge by introducing a score entropy objective, providing a theoretically more robust surrogate and establishing a connection between the concrete score and the continuous-time ELBO.

Although SEDD considers both uniform and absorbing transitions, masked diffusion (the absorbing case) exhibits significantly improved empirical performance. This approach introduces a [MASK] token representing an absorbing state and models the transitions between masked and unmasked states, analogous to the mechanism employed in masked language models. Recent work (Shi et al., 2024; Sahoo et al., 2024) further unifies the masked diffusion framework with continuous diffusion principles, resulting in simplified and theoretically grounded training and sampling procedures. This unification not only offers a more co-

herent understanding of masked diffusion models but also facilitates both theoretical and empirical progress through enhanced parameterization and engineering strategies. The present work primarily adopts this unified framework.

**LLM distillation**   Recent research on LLM distillation (Xu et al., 2024) focus on enhancing the efficiency and performance of smaller models while leveraging the strengths of larger ones. One of the main challenge lies in addressing the discrepancy betweeen training and inference. MiniLLM (Gu et al., 2024) proposes mixing teacher and student distributions to address training-inference mismatches, improving output quality and consistency. DistiLLM (Ko et al., 2024) builds on this by introducing a skew Kullback-Leibler divergence (KLD) loss to stabilize optimization and an adaptive off-policy strategy to enhance training efficiency, significantly reducing the computational burden associated with generating self-generated outputs. Hsieh et al. (2023) uses rationales generated by LLMs to train smaller, task-specific models effectively. This method highlights the importance of reasoning in distillation, allowing smaller models to achieve competitive performance even with limited data. Liu et al. (2024) explores a dynamic approach to distillation that integrates active learning techniques by iteratively selecting the most informative examples, thereby improving the efficiency and effectiveness of knowledge transfer from larger to smaller models. Fundamentally different from them, our distillation from the LLM to the diffusion model involves transferring knowledge from a unidirectional model to a bidirectional model. Nevertheless, we have discovered that certain techniques, like mitigating the distribution discrepancy between training and inference is helpful.

