# OpenReview forum: "Distilled Diffusion Language Models"
_ICLR.cc/2025/Conference — Submitted to ICLR 2025_

### Official Review · Reviewer_QecT · 2024-11-02

**Soundness:** 3
**Presentation:** 2
**Contribution:** 2
**Rating:** 3
**Confidence:** 4

**Summary:**

This paper proposes Target Concrete Score (TCS) to distill autoregressive language models to diffusion language models in order to enhance the latter one. The TCS method is applicable to a wide range of diffusion language models, both continuous and discrete ones. Comprehensive experiment supports the effectiveness of TCS.

**Strengths:**

1. The authors propose distillation from autoregressive models as an effective way to enhance the performance of diffusion language models.
2. The proposed method is theoretically grounded.
3. Empirical results show the effectiveness of the method in terms of perplexity.

**Weaknesses:**

1. The authors emphasize the error-correction ability of diffusion language models but do not show evidence to support it. Additionally, autoregressive models also have the potential to correct previous errors with chains of thought.
2. Although this paper narrows the performance gap between autoregressive and diffusion language models, diffusion language models still underperform autoregressive models in most tasks without unique advantages.
3. Insufficient experiments to study how the scales of teachers and students affect learning efficiency. It remains unclear whether the proposed methods help scale a diffusion language model.

**Questions:**

Do diffusion language models learned from scratch and learned with TCS distillation show similar patterns in intermediate generation steps?

---

### Official Review · Reviewer_X5hG · 2024-11-03

**Soundness:** 2
**Presentation:** 2
**Contribution:** 2
**Rating:** 3
**Confidence:** 4

**Summary:**

The paper introduces a novel framework for distilling knowledge from pre-trained autoregressive (AR) language models into non-autoregressive diffusion models. The core contribution is the Target Concrete Score (TCS) distillation, a method designed to bridge the gap between autoregressive and diffusion paradigms. It can apply to both discrete and continuous diffusion models and leverages any pre-trained autoregressive teacher model. Experiments on language modeling tasks show improvements in pre-trained diffusion language models and the ability to train new models from scratch.

**Strengths:**

1. The proposed method shows the potential to improve learning efficiency and perplexity, which is demonstrated through experiments on language modeling tasks.

**Weaknesses:**

1. **The method's introduction lacks systematic clarity**: the paper does not provide a comprehensive introduction to the process of distilling from autoregressive (AR) models to diffusion models. It is challenging to discern the specific difficulties involved in this distillation process and how the current work addresses these challenges.  It would benefit from a more structured explanation that highlights the novel contributions and breakthroughs of the proposed method.

2. **The experimental comparisons are insufficient**: the experimental section lacks a thorough comparison with existing diffusion models, particularly in terms of perplexity (PPL). While the paper presents baseline comparisons, it fails to include benchmarks against state-of-the-art (SOTA) diffusion models, which is crucial for validating the effectiveness of the proposed method. For instance, Table 3 follows the experimental setup of Ye et al., 2024, but does not include a comparative analysis of their results, limiting the ability to assess the method's performance.

3. **The validation of the method does not scale up in terms of model size and capability**: the paper does not sufficiently demonstrate the method's scalability, particularly in terms of distilling larger AR models. The ability to effectively distill knowledge from more complex AR models is crucial for validating the motivation behind transferring knowledge to diffusion models. However, the manuscript lacks discussions on whether the proposed method can scale up to handle larger models, which is a key aspect of assessing the practical viability of the approach.

**Questions:**

1. How does the proposed TCS distillation method compare to other state-of-the-art distillation techniques, especially in terms of efficiency and performance?
2. Could the authors provide more detailed experiments that systematically vary data size and model complexity to demonstrate the scalability of the proposed method?

---

### Official Review · Reviewer_5YQ6 · 2024-11-04

**Soundness:** 2
**Presentation:** 2
**Contribution:** 2
**Rating:** 5
**Confidence:** 2

**Summary:**

This paper explore the possibility of distilling a pre-trained autoregressive (AR) language model (teacher) into a non-autoregressive diffusion (non-AR) language model (student), combining the best of both worlds. The authors propose TCS distillation, a theoretically grounded framework that bridges autoregressive and diffusion paradigms, which can be broadly applicable to both discrete and continuous diffusion models, with any pre-trained autoregressive teacher model.

**Strengths:**

- Knowledge distillation is a potential direction to enhance diffusion models.
- The results are good.
- This paper is well writen.

**Weaknesses:**

- I am not sure whether several numbers in Table 1 is missing.

**Questions:**

see weaknesses.

---

### Official Review · Reviewer_iLT2 · 2024-11-05

**Soundness:** 3
**Presentation:** 2
**Contribution:** 2
**Rating:** 3
**Confidence:** 4

**Summary:**

This manuscript introduce Distilled Diffusion Language Models (DDLM), a framework to distill pre-trained autoregressive (AR) language models into denoising diffusion language models.
A key contribution is the Target Concrete Score (TCS) distillation objective, aiming to bridge the gap between AR and diffusion models. Specially, top-K and gradient-informed estimation are proposed to efficiently estimate the TCS.
DDLM is evaluated for both discrete and continuous Diffusion LMs, on several language modeling and reasoning tasks, showing its effectiveness in improved performance of Diffusion LMs with faster parallel generation.

**Strengths:**

The paper presents a new way to distill AR language models into diffusion language models to bridge their performance gap, where the TCS distillation objective effectively connects different types of models.  The paper's evaluation on a range of tasks demonstrates its improved performance on complex tasks like in-filling and arithmetic. Moreover, the potential for faster parallel generation is also a advantage over autoregressive counterparts.

**Weaknesses:**

I think one of the major weaknesses lies in evaluation on more and widely-used standard LLM benchmark.
The authors only evaluate the models against PPL where PPL/likelihood of Diffusion LMs are often not exact and cannot serve as a standalone indicator for real language capabilities. Therefore, the paper should provide more detailed comparisons of DDLM with existing AR-LMs (e.g, LLAMA-3 8B) on downstream language tasks beyond GSM8K-Aug, such as BBH, MMLU, multilingual tasks like translation, etc. Plus, case study of samples generated by DDLM is needed to assess the behaviors of the model, especially for reasoning tasks.
ALL of these are important to convincingly demonstrate the proposed framework's ability to generalize across a wider range of language tasks and datasets.

Moreover, despite the great promise of self-correction and bidirectional context in Diffusion LMs, AR-LMs can achieve similar results through reflection or complicated chain-of-thought reasoning, as demonstrated by O1. Additionally, open-ended reasoning is particularly challenging for Diffusion LMs because they require pre-specified sequence lengths. Faster parallel generation is good, but AR-LLMs enjoy many MLSYS optimizations thanks to exactly their autoregressive/recursive decomposition especially at inference time.
At the end of the days, what is the real potential of developing Diffusion LMs for natural language tasks, as an alternative for AR-LLMs? And to reach this goal, what major challenges need to be addressed?

**Questions:**

See weaknesses. Will consider adjusting my initial rating upon authors' responses.

---

### Meta-Review · Area_Chair_ENFy · 2024-12-19

**Metareview:**

This paper proposes a novel framework DDLM that distills pre-trained autoregressive (AR) language models into diffusion language models by utilizing a Target Concrete Score (TCS) distillation objective. The core idea is to integrate the strengths of both AR-LMs and diffusion models, aiming to bridge their performance gap. The authors focus on improving the efficiency and effectiveness of diffusion language models through this distillation process, which is theoretically grounded and applicable to both discrete and continuous diffusion models.

Pros:
1. The paper addresses an interesting and emerging area in NLP by attempting to enhance the performance of diffusion-based language models using knowledge distillation from AR models. The topic itself is an appealing proposition for niche exploration within the linguistic model communities, marrying two paradigms often seen in isolation.


Cons:
1. There is a need for broader evaluations across more standard and robust language model benchmarks. Reviewers have noted insufficient comparative analyses with existing state-of-the-art diffusion and autoregressive models, which are crucial for demonstrating efficacy and practical viability. Authors should include thorough experiments to compare against benchmarks like MMLU, multilingual tasks, and others mentioned by reviewers to substantiate claims of generalization.

2. The methodology's presentation is noted as lacking in cohesion and clarity. A more detailed and systematic explanation of the TCS distillation process and its novel contributions compared to existing works is needed, particularly addressing distillation from AR models to diffusion models. Providing a structured narrative on the challenges and breakthroughs can greatly enhance understanding and uptake.

3. There is a deficiency in demonstrating the scalability of the proposed method in terms of handling larger models and comprehending the error-correction capabilities of diffusion models, particularly compared against AR models’ chain-of-thought reasoning. Addressing these aspects with detailed discussions and experiments can provide richer insights into DDLM’s potential and shortcomings.


Suggestions:
1. Reviewers highlighted the necessity of providing a deeper exploration into how diffusion models evolve across intermediate generation steps when using TCS distillation. Such insights could be beneficial for understanding the dynamics of the trained models better.

2. Some inconsistencies and potential missing data (like in Table 1) were noted, which needs to be revisited for clarity and completeness.

**Additional Comments On Reviewer Discussion:**

There was no rebuttal.

---

### Decision · Program_Chairs · 2025-01-22

Reject